# PeerJ

# Effect of obesity and exercise on the expression of the novel myokines, Myonectin and Fibronectin type III domain containing 5

Jonathan M. Peterson[1,2], Ryan Mart[2] and Cherie E. Bond[3]

[1] Department of Health Sciences, College of Public Health, East Tennessee State University, Johnson City, TN, USA
[2] Quillen College of Medicine, East Tennessee State University, Johnson City, TN, USA
[3] Ferrum College, Ferrum, VA, USA

Corresponding author
Jonathan M. Peterson,
petersonjm1@etsu.edu

## ABSTRACT

Metabolic dysfunction in skeletal muscle is a major contributor to the development of type 2 diabetes. Endurance exercise training has long been established as an effective means to directly restore skeletal muscle glucose and lipid uptake and metabolism. However, in addition to the direct effects of skeletal muscle on glucose and lipids, there is renewed interest in the ability of skeletal muscle to coordinate metabolic activity of other tissues, such as adipose tissue and liver. The purpose of this study was to examine the effects of endurance exercise on the expression level of two novel muscle-derived secreted factors, or myokines, Myonectin and Fibronectin type III domain containing 5 (FNDC5), the precursor for Irisin.

**Methods.** We performed immunoblot analysis and quantitative real-time PCR analysis of Myonectin and FNDC5 in the diaphragm muscles of obese Zucker rat (OZR) and lean Zucker rat (LZR) with 9 weeks of aerobic training on a motorized treadmill.

**Results.** We show that myonectin gene expression is increased in the OZR model of obesity and decreases with exercise in both lean and obese Zucker rats. Conversely, myonectin protein concentration was elevated with exercise. Similarly, FNDC5 mRNA levels are significantly higher in the OZR, however exercise training had no effect on the expression level of FNDC5 in either the LZR or OZR. We did not observe any difference in muscle protein content of Irisin with obesity or exercise.

**Conclusion.** Our data shows that exercise training does not increase either FNDC5 or myonectin gene expression, indicating that increased transcriptional regulation of these myokines is not induced by exercise. However, our data also indicates a yet to be explored disconnect between myonectin gene expression and protein content. Further, this report highlights the importance of verifying reference genes when completing gene expression analysis. We found that many commonly used reference genes varied significantly by obesity and/or exercise and would have skewed the results of this study if used to normalize gene expression data. The unstable reference genes include: beta-Actin, beta-2-microglobulin, Non-POU domain containing, octamer-binding, Peptidylprolyl isomerase H, 18S ribosomal RNA, TATA box binding protein and Transferrin receptor.

Obesity and diabetes are the top health problems in the developed world, and major contributors to the development of cardiovascular disease (*Ng et al., 2013*). Skeletal muscle metabolism is an important regulator in control of whole body glucose and lipid homeostasis. Further, the reduction in insulin-mediated skeletal muscle glucose uptake has long been recognized to be an important underlying mechanism of type 2 diabetes (*Bogardus, 1989*). Lifestyle modification, specifically increased physical activity, has demonstrated enormous therapeutic potential to reverse skeletal muscle insulin resistance.

While the direct role of skeletal muscle metabolism in regulating glucose and lipid metabolism is well established, the potential endocrine-like functions of skeletal muscle to influence glucose and lipid metabolism in other tissues have only recently begun to be investigated. With the advent of proteomics, a number of muscle-derived secreted factors, collectively called myokines, have been identified (*Norheim et al., 2011*; *Henningsen et al., 2010*). Interleukin 6 (IL-6) is the first and most well studied myokine, and it is increased with exercise (*Northoff & Berg, 1991*; *Haahr et al., 1991*; *Ostrowski et al., 2001*). IL-6 acts to stimulate hepatic glucose production and enhances glucose uptake by insulin-sensitive tissues (*Stouthard et al., 1995*; *Stouthard, Oude Elferink & Sauerwein, 1996*). These studies have provided the first endocrine-like function of skeletal muscle and established a link between exercise and systemic metabolic parameters (*Sandler et al., 1990*). In addition to IL-6, a large number of other muscle-derived secretory proteins have been identified. The purpose of this paper was to examine the impact of obesity and chronic exercise training on two of these novel myokines: (1) fibronectin type III domain containing 5 (FNDC5), and (2) C1q TNF related Protein 15/myonectin, hereafter referred to as myonectin. FNDC5 is a membrane protein that is cleaved and its proteolytic cleavage product is secreted as the hormone, irisin (*Bostrom et al., 2012*). Myonectin is a newly discovered protein with a characteristic C1q domain sequence shared by proteins within the novel CTRP protein family (*Seldin et al., 2012*).

Both myonectin and irisin have shown promise as therapeutic targets for metabolic diseases known to improve with exercise. Myonectin lowers circulating levels of free fatty acids by increasing uptake in adipose and liver tissues, and is increased with exercise, but lowered in a high fat diet model of obesity (*Seldin et al., 2012*). Irisin, on the other hand, increases energy expenditure by inducing brown-fat-like conversion of white adipose tissue (*Bostrom et al., 2012*). However, the regulation of circulating irisin levels and FNDC5 gene expression by obesity and exercise are unclear and recently reviewed by *Polyzos et al. (2013)*. Briefly, exercise causes an increase circulating irisin protein levels and/or FNDC5 mRNA expression in some (*Bostrom et al., 2012*; *Huh et al., 2012*; *Brenmoehl et al., 2014*), but not all studies (*Huh et al., 2012*; *Kurdiova et al., 2014*; *Timmons et al., 2012*; *Timmons et al., 2005*; *Gallagher et al., 2010*; *Seo et al., 2014*), whereas, obesity

has either a positive (*Polyzos et al., 2013*; *Huh et al., 2012*; *Timmons et al., 2012*) or a negative (*Seo et al., 2014*; *Peterson et al., 2008b*) association with irisin/FNDC5.

It is the purpose of this study to examine the combined effects of exercise and obesity on the regulation of myonectin and FNDC5 gene expression. This study may give clues to understanding the mechanism behind the endocrine benefits of regular exercise with obesity.

## METHODS

### Animals

The diaphragm muscles were kindly provided by the lab of Stephen E. Alway, as reported from a previous study (*Peterson et al., 2008b*; *Peterson et al., 2008a*). Briefly, equal numbers of 6-wk-old male Obese Zucker rats (OZR) and lean zucker rats (LZR) (Harlan, Indianapolis, IN) were randomly assigned to control (Control, $n = 8$) or training (Exercised, $n = 8$) groups. The OZR is a genetic model of obesity due to the presence of the recessive missense mutation (*fa/fa*) in the leptin receptor gene, whereas the LZR has a functioning leptin receptor (Fa/fa or Fa/Fa) (*Ogawa et al., 1995*; *Stern et al., 1972*; *Godbole & York, 1978*; *Takaya et al., 1996*). Compared to the LZR, the OZR exhibits severe obesity, hyperphagia, hyperinsulinemia, hyperleptinemia, and hyperlipidemia (*Ogawa et al., 1995*; *Stern et al., 1972*; *Godbole & York, 1978*; *Takaya et al., 1996*). Animals were housed in pathogen-free conditions, two per cage, at 20–22 °C with a reversed 12:12-h light–dark cycle, and fed rat chow and water ad libitum throughout the study period. All animal procedures were conducted in accordance with institutional guidelines, and ethical approval was obtained from the Animal Care and Use Committee at West Virginia University (ACUC #07-0302).

### Training protocol

LZR and OZR were exercise trained by running on a level motorized rodent treadmill (Columbus Instruments, Columbus, OH) 5 d/wk for 9 wk, as previously reported (*Peterson et al., 2008b*; *Peterson et al., 2008a*). Briefly, during the first 4 wk, the speed of the treadmill and duration of the training sessions was increased gradually from a speed of 10 m/min for 10 min to a final running speed of 20 m/min for the OZR and 24 m/min for the LZR. During the training sessions, mild electrical shock was applied, if necessary, to maintain the animals' running motivation. A slower final running speed was used in the OZR group to compensate for the increased intensity of exercise for these animals that resulted from their greater body weight as compared with LZR. As reported previously, the workload was estimated based on the following formula: Work = 1/2 mass * velocity squared ($W = 1/2 \, m * V^2$). The average body weight of 500 g for the OZR and 350 g for the LZR were used to calculate the treadmill speeds to produce an approximate work output of 0.028 J. These speeds were also reliably maintained by the OZR and LZR, with minimal requirements for external motivation by the investigators. This approach was successful as determined by similar increases in mitochondrial protein content and activity in the trained groups (*Peterson et al., 2008b*; *Peterson et al., 2008a*). Animals assigned to the

control group were handled daily and exposed to the noise of the running treadmill by placing their cages next to the treadmill during the exercise session.

## Tissue collection

Forty-eight hours after the last training session and an overnight fast ($\sim$16 h), the animals were anesthetized with injections of pentobarbital sodium (50 mg/kg ip) and euthanized via cardiac puncture. The diaphragm muscles were quickly removed, frozen immediately in liquid nitrogen and stored at $-80\,^{\circ}$C until further analysis. It has been previously documented that exercise training has similar effects on the diaphragm as leg muscles (*Moore & Gollnick, 1982*). Further, we confirmed the expected increase in the mitochondrial protein Cytochrome c Oxidase Subunit IV (COX IV) as a marker of total mitochondrial content.

## RNA isolation and reverse transcription

Total RNA was extracted according to standard procedures. Briefly, tissues were homogenized in Trizol reagent (Life Technologies) using a Kinematica polytron in three 30 s bursts, separated by 10 min incubations on ice. After centrifugation at 13.2 rcf, 4 $^{\circ}$C for 5 min to remove residual particulates, phase separation was accomplished using 3-bromo-5-chloropentane, followed by centrifugation for 15 min at 13.2 rcf, 4 $^{\circ}$C. RNA was precipitated from the aqueous phase by mixing with an equal volume 70% ethanol, and then was loaded onto a nucleotide binding column (RNeasy Mini-Kit; Qiagen). On-column DNA digestion was performed using RNase-free DNase (Qiagen) to eliminate residual genomic DNA contamination as per the manufacturer's instructions. RNA was eluted in 50 µl RNase-free water; purity (RIN $\geq$ 7.0) and concentration were confirmed by microfluidic capillary electrophoresis using an Agilent BioAnalyzer. 1 µg total RNA of each sample was reverse transcribed in a final volume of 20 µl, using GoScript® cDNA synthesis reagents (Promega).

## Analysis of reference genes

To screen for potential stable reference genes, an aliquot 1 µl of prepared cDNA from each animal was pooled by group and treatment and the relative content of each reference gene was determined by PCR array according to manufactures directions (RT$^2$ Profiler PCR Array; Rat Housekeeping Genes; Qiagen PARN-000ZA). Assuming a perfect efficient reaction, the difference between 1 quantification cycle (Cq) equals a 2-fold difference in starting RNA quantity. Variability of reference genes was deemed to be unacceptable if the maximum difference among the four groups was greater than 0.5 Cq. Reference genes examined are listed in Table 2.

## Quantitative real time PCR analysis

Validated PCR primers for Myonectin, FNDC5/irisin, Hprt1, Ldha, and RN18S were purchased from SABiosciences (Table 3). A 10-fold dilution series of DNA amplicons generated from an untrained LZR rat muscle was employed as a standard curve for each gene of interest, and the qPCR efficiency was determined for each gene, using a Bio-Rad

Cfx thermocycler. Briefly, 0.5 μl of cDNA from the reverse transcription reaction was incubated in appropriate mix (SABbiosciences) for an initial denaturation at 94 °C for 30 s, followed by 40 PCR cycles each consisting of 95 °C for 0 s, 61 °C for 7 s, and 72 °C for 10 s. All qRT-PCR primers displayed a coefficient of correlation greater than 0.99 and efficiencies between 90% and 110%. Data is reported as copy number per amount of starting RNA (0.25 ng per reaction). Specificity of amplification products was further confirmed by analyzing melting curve profiles for primers and products and subjecting the amplification products to agarose gel electrophoresis.

## Immunoblot analysis

Diaphragm muscles were prepared in lysis buffer (20 mM Tris pH 7.5, 150 mM NaCl, 1% Nonidet P-40, 0.5% sodium deoxycholate, 1 mM EDTA, 0.1% SDS) with protease and phosphatase inhibitor cocktails (Sigma). The protein concentration was determined using a Coomassie Plus protein assay reagent (Thermo Scientific). For each sample, 10 μg of protein were loaded and separated on a SDS-polyacrylamide gel, according manufactures direction (BioRad). The proteins were then transferred to Nitrocelluloseous membranes blocked with milk and incubated with Rabbit polyclonal anti-peptide antibody that can recognize myonectin (epitope 77-KQSDKGI NSKRRSKARR-93). Myonectin antibody was kindly provided by the lab of GW Wong and had been used previously (*Seldin et al., 2012*). Glyceraldehyde 3-phosphate dehydrogenase (GAPDH) antibody was purchased from Novus Biologicals (NB300-221), COX IV antibody was purchased from Cell Signaling Technology (4844), and FNDC5 antibody was purchased from Abcam (ab131390). Antibody detection was performed with the appropriate horseradish perioxidase-conjugated secondary antibodies. Visualizations were completed with MultiImage III FuorChem® M (Alpha Innotech) and quantifications were performed by Alphaview Software (Alpha Innotech).

## Statistical analysis

Analyses were performed using GraphPad Prism® 5 software package. Student's *T*-test was used for comparisons between control and exercise trained animals within the same genotype (Fig. 3). A one-way ANOVA, followed by Bonferroni multiple comparisons post hoc analysis, were performed when comparisons were made among all groups (Fig. 2). Statistical significance was accepted at $P < 0.05$. All data are given as means $\pm$ SE. No statistical analysis was performed on the PCR array data of pooled samples (Fig. 1). A reference gene was deemed acceptable for further analysis if the maximum Cq difference among the pooled samples from the 4 groups was less than 0.5.

## RESULTS AND DISCUSSION

The purpose of this study was to examine the regulation of myonectin and FNDC5 in skeletal muscles of a genetic model of obesity and then to determine the combined effects of obesity and exercise on the gene expression of these two proteins. To our knowledge, this is the first study to examine the effect of obesity combined with exercise training on skeletal muscle gene expression of the novel myokines myonectin and FNDC5. Although a number

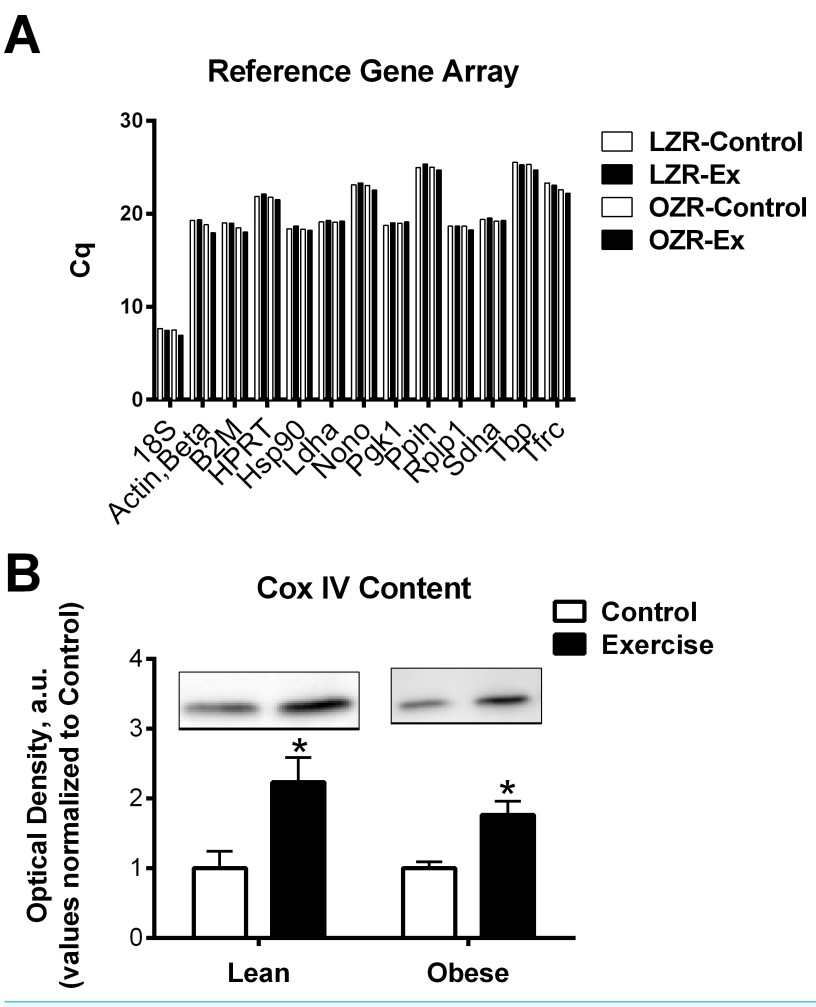

**Figure 1 Reference genes and mitochondria protein.** (A) To screen reference genes for relative stability pooled cDNA for each group was examined to determine the Cq number by PCR array (RT$^2$ Profiler PCR Array; Rat Housekeeping Genes; Qiagen PARN-000ZA). (B) Mitochondrial protein COX IV was measured in the Lean and obese animals as a marker of total mitochondrial content. Representative blots are show. The data are expressed in arbitrary units with values normalized to mean control value within phenotype. **Abbreviations**: Cq, quantification cycle; LZR, Lean Zucker Rat; OZR, Obese Zucker Rat; ET, Exercise trained; Actb, Actin, beta; B2m, Beta-2 microglobulin; Hprt1, Hypoxanthine phosphori­bosyltransferase 1; Hsp90, Heat shock protein 90 alpha (cytosolic), class B member 1; Ldha, Lactate dehydrogenase A; Nono, Non-POU domain containing, octamer-binding; Pgk1, Phosphoglycerate kinase 1; Ppih, Peptidylprolyl isomerase H (cyclophilin H); Rplp1, Ribosomal protein, large, P1; Sdha, Succinate dehydrogenase complex, subunit A, flavoprotein (Fp); Tbp, TATA box binding protein; Tfrc, Transferrin receptor; COX IV, Cytochrome c Oxidase Subunit IV.

of cross sectional studies have compared irisin/FNDC5 levels to body mass index (BMI) the results have been contradictory (*Huh et al., 2012*; *Timmons et al., 2012*; *Timmons et al., 2005*; *Gallagher et al., 2010*; *Choi et al., 2013*; *Stengel et al., 2013*; *Moreno-Navarrete et al., 2013*; *Kurdiova et al., 2014*). Based on the previous literature, we initially hypothesized that the gene expression of both of these proteins would be reduced with obesity and that the levels would increase with endurance exercise (*Bostrom et al., 2012*; *Seldin et al., 2012*).

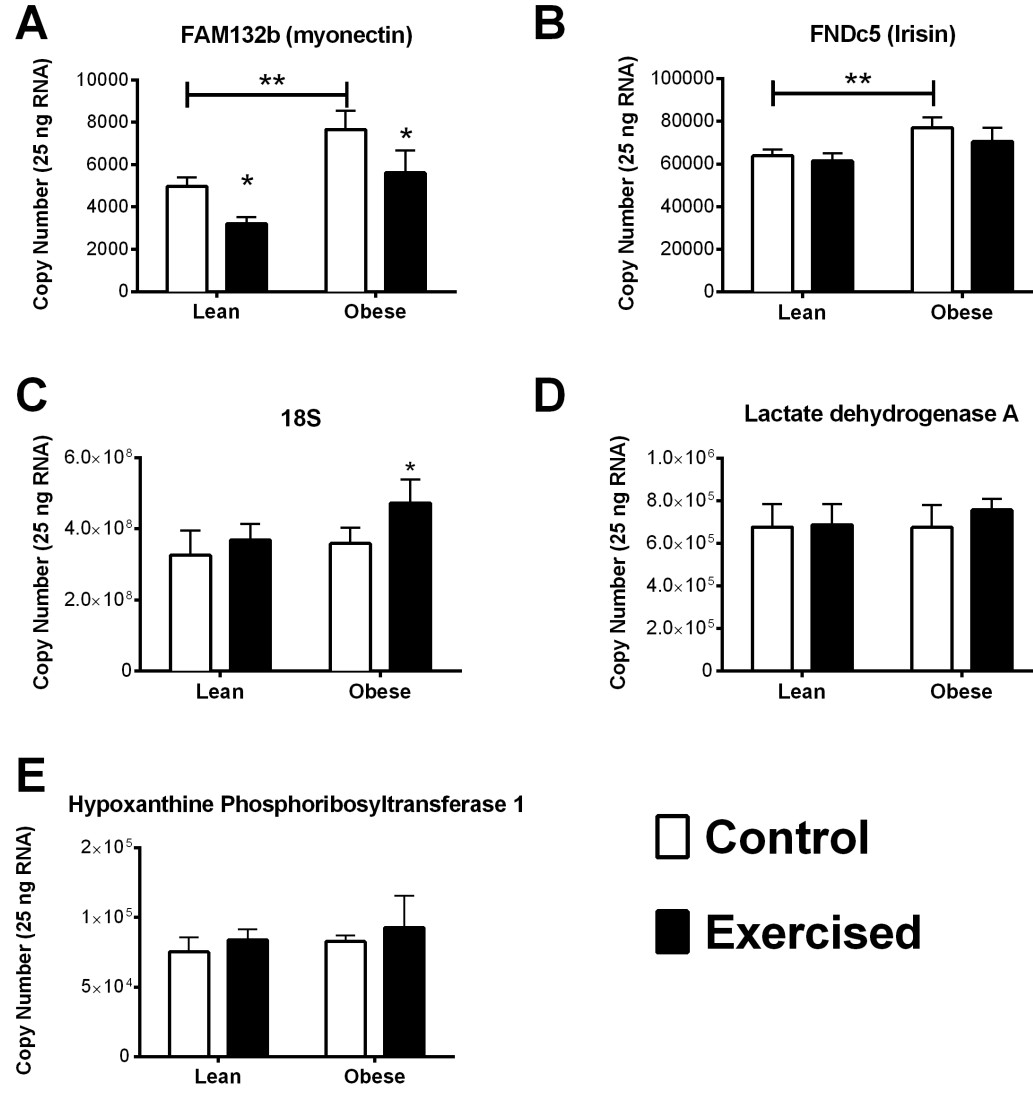

**Figure 2 Quantitative RNA analysis.** Myonectin (A), FNDC5/irisin (B), RN18S (C), Ldha (D), and Hprt1 (E). Validated PCR primers were purchased from SABiosciences (Table 3). A standard curve was generated from 10-fold dilution series of DNA amplicons for each gene of interest. All qRT-PCR primers displayed a coefficient of correlation greater than 0.99 and efficiencies between 90% and 110%. Data is reported as copy number per amount of starting RNA. The main effects of obesity (OZR $\times$ LZR) and exercise interaction (obesity $\times$ exercise) in these animals were analyzed by a two-way ANOVA. Data are presented as means $\pm$ SE. *$p < 0.05$, data significantly different between control and exercised groups. **$p < 0.05$, data from OZR animals was significantly different from the LZR animals.

## Characterization of animals

Our goal in the training regime was to attempt to match final workload between the LZR and OZR. This approach was successful as determined by similar increases in mitochondrial protein content and activity in the trained groups (*Peterson et al., 2008b*; *Peterson et al., 2008a*) and Fig. 1B. Although this workload was sufficient to lower body

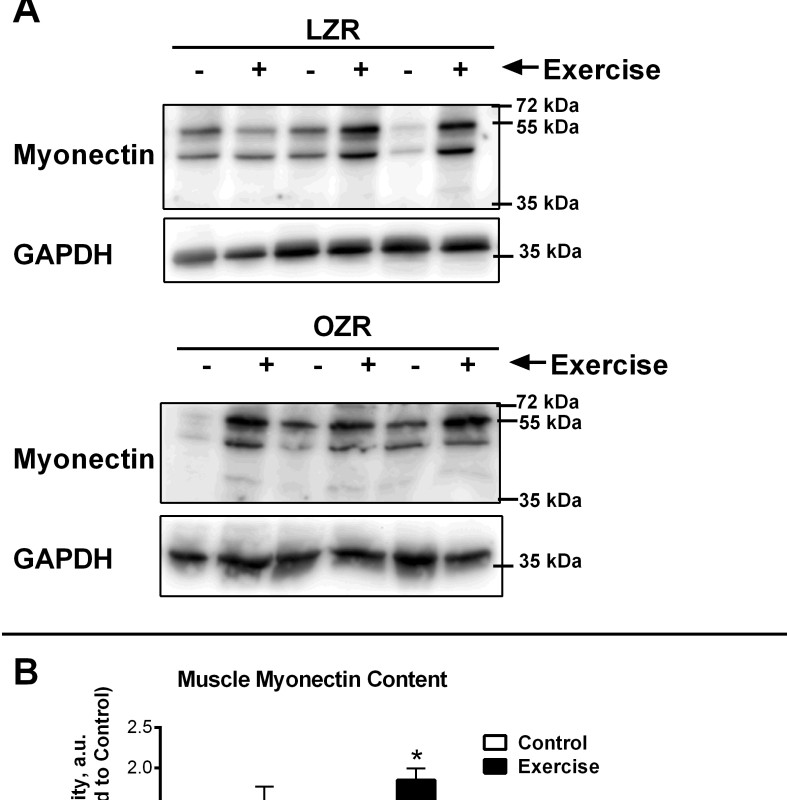

**Figure 3 Relative Myonectin/CTRP15 content.** Relative Myonectin/CTRP15 content was examined in the rat diaphragm muscle of the male lean zucker (LZR) and obese zucker rats (OZR). Exercised animals were trained on a motorized treadmill for 9 wk. Control animals were exposed to the similar environment (positioned next to the treadmill) but were not exercised. (A) Shows representative western blots for Myonectin and GAPDH. (B) The data are expressed in arbitrary units with values normalized to mean control value within phenotype.

weight and fasting insulin levels (Table 1) in the OZR, it was not sufficient to induced significant changes in the LZR in these variables.

## Identification of appropriate reference genes

Our data was able to confirm that HPRT, HSP90, Ldha, Pgk1, Rplp1, and Sdha remained relatively stable (Cq variability less than 0.5) regardless of obesity or exercise training (Fig. 1A). However, we also observed that there was greater than 1 Cq difference, among the groups examined, in gene expression of Actb, B2m, and Tfrc (Fig. 1A and Table 2). Assuming an efficient reaction, 1 Cq difference represents an approximate 2-fold difference in starting RNA content. This indicates that some commonly used reference genes are effected by the specific set of conditions described in this study and therefore are

Table 1 **Baseline characteristics of study animals.** Values are means SE.

| | LZR-SED ($n = 8$) | LZR-EX ($n = 8$) | OZR-SED ($n = 8$) | OZR-EX ($n = 8$) |
|---|---|---|---|---|
| Mass, g | 358 ± 18 | 360 ± 19 | 685 ± 24.5 | 502 ± 24[*] |
| Blood glucose, mg/dl | 116 ± 9 | 108 ± 10 | 188 ± 24 | 179 ± 41[*] |
| Plasma insulin, ng/ml | 1.5 ± 0.4 | 1.3 ± 0.5 | 10.7 ± 1.2 | 6.7 ± 1.6 |

**Notes.**

$n$, no. of animals; LZR, lean Zucker rat; EX, exercise; SED, non-exercised; OZR, obese Zucker rat.

[*] $P + 0.05$ vs. Sed.

Table 2 **Rat housekeeping genes.** Variability of reference genes was deemed to be unacceptable if the maximum difference among the four groups was greater than 0.5 Cq. Reference genes examined are listed in Table 1.

| Gene name | Abbreviation | Maximum Cq difference |
|---|---|---|
| Actin, beta | Actb | 1.4 |
| Beta-2 microglobulin | B2m | 1.0 |
| Hypoxanthine phosphoribosyltransferase 1 | Hprt1 | .5 |
| Heat shock protein 90 alpha | Hsp90 | .49 |
| Lactate dehydrogenase A | Ldha | .19 |
| Non-POU domain containing | Nono | .78 |
| Phosphoglycerate kinase 1 | Pgk1 | .36 |
| Peptidylprolyl isomerase H | Ppih | .61 |
| RPLP1 ribosomal protein, large, P1 | Rplp1 | .44 |
| Succinate dehydrogenase complex, subunit A | Sdha | .33 |
| TATA box binding protein | Tbp | .85 |
| Transferrin receptor | Tfrc | 1.13 |

**Notes.**

Cq, quantification cycle.

Table 3 **Quantitative real time PCR analysis.** Validated PCR primers for Myonectin, FNDC5/irisin, Hprt1, Ldha, and RN18S were purchased from SABiosciences.

| Gene name | Abbreviation | Accession # | Catalog number |
|---|---|---|---|
| Hypoxanthine phosphoribosyltransferase 1 | Hprt1 | NM_012583.2 | PPR42247F |
| 18S ribosomal RNA | RN18S | NR_046237.1 | PPR72042A |
| Lactate dehydrogenase A | Ldha | NM_017025 | PPR56603 |
| Myonectin; C1q TNF Related Protein 15; Family with sequence similarity 132, member B | Myonectin | XM_001060107.2 | PPR68386A |
| Fibronectin type III domain-containing protein 5; also known as irisin | Irisin | XM_001060505.2 | PPR46702A |

inappropriate to use as reference genes, normalizing factors that control for equal input of total RNA when performing relative gene expression analyses. Further, these data highlight the importance of exploring the stability of reference genes when performing qPCR analysis. It is possible that some of the conflicting data regarding the regulation of FNDC5 mRNA could be due to artifacts created by unreliable reference genes. Only one of the studies cited in this manuscript confirmed reference gene expression was unchanged between groups (*Gallagher et al., 2010*). Further, the two most commonly used reference genes were 18S ribosomal RNA (RN18S) and Actb, both of which exhibited excessive variability under experimental conditions, prohibiting their use as reference genes in this model.

Of the stable reference genes, we performed quantitative qPCR analysis of HPRT and Ldha to use as reference genes in our analysis. Quantitative qPCR showed that there was no significant difference between the starting copy number of HPRT and Ldha among the groups in our analysis (Figs. 2D and 2E). Interestingly, the reference gene RN18S was also examined by quantitative PCR analysis and found to be significantly elevated in the obese exercised group compared to all other groups.

## Effect of obesity on myonectin and FNDC5

It has been previously documented that myonectin protein and mRNA levels are downregulated after 12 weeks of high fat diet-induced obesity. Whereas, the data concerning the relationship of FNDC5/Irisin with obesity and type 2 diabetes is varied (*Huh et al., 2012*; *Timmons et al., 2012*; *Timmons et al., 2005*; *Gallagher et al., 2010*; *Seo et al., 2014*; *Kurdiova et al., 2014*). Nevertheless, we found that the OZR had significantly higher expression levels of both myonectin and FNDC5 compared with the LZR (Figs. 2A and 2B). There are a number of possibilities to explain this data. The first is that leptin plays a role in the regulation of both FNDC5 and myonectin. Obesity occurs in the OZR model due to a nonfunctioning leptin receptor. Therefore, any leptin-mediated regulation which occurs in the high fat diet-induced model of obesity would be absent in our model. This also raises the possibility that leptin signaling may be a contributing factor in the inconsistency regarding the relationship between BMI and FNDC5/irisin, as leptin levels can vary significantly among person with a BMI less than 30 (*Horn et al., 1996*; *Scholz et al., 1996*). Previous analyses of the association between BMI and FNDC5/irisin were across the entire spectrum of BMI (*Huh et al., 2012*; *Timmons et al., 2012*; *Timmons et al., 2005*; *Choi et al., 2013*; *Moreno-Navarrete et al., 2013*; *Kurdiova et al., 2014*). Another possibility, in regards to the discrepancy with myonectin data, is that a diet high in fat may induce the expression myonectin regardless of obesity. Previous work has demonstrated that mice challenged with a single dose of emulsified intralipid will show an approximate 500% increase in circulating myonectin levels (*Seldin et al., 2012*). In the OZR model of obesity, animals are fed a chow diet and become obese due to excessive caloric intake (*Ogawa et al., 1995*; *Stern et al., 1972*; *Godbole & York, 1978*; *Takaya et al., 1996*). Nevertheless, this finding may be serendipitous as these myokines have yet to be examined in a model with dysfunctional leptin signaling. These observations deserve more extensive analysis than

was possible within the scope of the current study. A third and unfortunate possibility is that the reference genes by which previous studies normalized the gene expression were affected by the study parameters and erroneously skewed the gene expression data.

Additionally, we attempted to measure irisin levels in the muscles samples using a commercially available antibody (Anti-FNDC5 antibody; ab131390). However, according to the manufacturers there should be a band at 22 kDa and an unidentified band at ~45 kDa (which corresponds to the size of FNDC5, but was unable to be confirmed within the constraints of this study). Although we successfully detected the band at 45 kDa, we only detected a faint band at 22 kDa in some of the samples. Regardless there were no differences observed among the groups examined of either the 45 kDa or 22 kDa band (data not shown). It is highly likely that differences may still be detected in circulating irisin levels, however as this study was a re-examination of previously acquired muscle samples the serum is no longer available from the animals studied.

## The combined effects of Obesity and exercise training on Myonectin expression

It has been previously documented that short-term exercise exposure (3-weeks free wheel running) increases the gene expression of myonectin (*Seldin et al., 2012*). Although myonectin expression had not been examined after long-term exercise exposure, we were surprised to find that chronic exercise (9-weeks) reduced myonectin expression regardless of obesity status (Fig. 2A). Because our findings were contrary to what we anticipated, we also examined the muscle protein content of myonectin (Fig. 3). These results were even more confounding since, although myonectin gene expression was reduced with exercise, myonectin protein content was elevated with exercise, regardless of obesity. Unfortunately, serum samples were no longer available from these animals to determine the changes to circulating myonectin levels. However, these data suggest that myonectin may act in an autocrine fashion to regulated it's own expression, as has been reported recently for irisin (*Vaughan et al., 2014*) and for other myokines such as IL-6 (*Bustamante et al., 2014*). This potential autocrine regulation warrants further analysis but was beyond the confines of the current study.

## The combined effects of Obesity and exercise training on FNDC5 gene expression

According to the literature neither short-term nor chronic exercise alters FNDC5 gene expression (*Bostrom et al., 2012*; *Brenmoehl et al., 2014*). FNDC5 is the precursor for Irisin, and it has been suggested that exercise causes cleavage of FNDC5, releasing irisin and driving the exercise-induced 'browning' of white adipose tissue (*Bostrom et al., 2012*; *Castillo-Quan, 2012*). This indicates that FNDC5 levels may not be directly regulated by exercise. However, because FNDC5 levels are reduced with obesity and insulin resistance we expected to see a restoration of FNDC5 levels with exercise training in the insulin resistant obese OZR. Although we confirmed that exercise did not effect FNDC5 gene expression, contrary to our initial hypothesis we found that FNDC5 mRNA levels were

elevated in the OZR (Fig. 3B). As stated earlier, this finding indicates that FNDC5 may be regulated by leptin, or by dietary fat content.

## Conclusion

Both myonectin and irisin have been linked to improved metabolic health outcomes. Myonectin coordinates lipid homeostasis in liver and adipose tissue with the metabolic demands of skeletal muscles (*Seldin et al., 2012*), whereas, irisin increases energy expenditure in mice through the browning of white adipose tissue (*Bostrom et al., 2012*). To date, the combined effects of exercise and obesity on the regulation of myonectin and irisin have not been examined. This study shows that in the OZR both myonectin and FNDC5 gene expression are elevated. Further, contrary to previous findings, myonectin gene expression was negatively regulated by exercise, regardless of obesity. The findings of this study implicate leptin signaling and high fat diet as a potential novel mechanism in the regulation of these proteins, and these possibilities warrant future study. Unfortunately, serum samples were no longer available to analyze the combined effect of exercise and obesity on the circulating levels of these novel myokines. Future studies should consider examining the circulating levels of these myokines in a leptin deficient model with and without exercise or treatment with recombinant leptin.

## ACKNOWLEDGEMENTS

We would like to thank the East Tennessee State University Molecular Biology Core Facility for support in completing quantitative Real-time PCR experiments. We would like to thank the Lab of Stephen E. Alway for providing the diaphragm muscles used in this study.

### Funding

This work was funded by a grant from the East Tennessee State University Research Development Committee Major Grants Program (E83044). The funders had no role in study design, data collection and analysis, decision to publish, or preparation of the manuscript.

### Grant Disclosures

The following grant information was disclosed by the authors:
East Tennessee State University Research Development Committee Major Grants Program: E83044.

### Competing Interests

The authors declare there are no competing interests.

### Author Contributions

- Jonathan M. Peterson conceived and designed the experiments, performed the experiments, analyzed the data, contributed reagents/materials/analysis tools, wrote the paper, prepared figures and/or tables, reviewed drafts of the paper.

**Peer**J ___________________________________________________________________

- Ryan Mart performed the experiments, reviewed drafts of the paper.
- Cherie E. Bond performed the experiments, analyzed the data, contributed reagents/materials/analysis tools, reviewed drafts of the paper.

## Animal Ethics

The following information was supplied relating to ethical approvals (i.e., approving body and any reference numbers):

All animal procedures were conducted in accordance with institutional guidelines, and ethical approval was obtained from the Animal Care and Use Committee at West Virginia University (ACUC# 07-0302).

## Supplemental Information

Supplemental information for this article can be found online at http://dx.doi.org/ 10.7717/peerj.605#supplemental-information.

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
