# Peer review of "Effect of obesity and exercise on the expression of the novel myokines, Myonectin and Fibronectin type III domain containing 5"

_PeerJ, doi:10.7717/peerj.605_

## Round 0.1 · original submission · Major Revisions

Dear Dr. Cui:

Thank you for submitting your manuscript to Peerj. After careful consideration, we feel that it has merit, but is not yet suitable for publication as it currently stands. Therefore, my decision is "Major Revision."

Reviewer 1 ·

Basic reporting

No Comments

Experimental design

Statistical Page 5, line 181-188- The statistical methods read “A one‐way ANOVA was performed when comparisons were made among all groups (figure 2)” but there is no mention of post-hoc test to control for pair-wise error. The authors should mention which test was performed to control for this (Bonferroni’s, Tukey’s, etc). If not previously performed, one of these techniques should be used prior to publication.

Validity of the findings

Please see “Experimental Design” and “General Comments for the Author”.

Additional comments

The manuscript entitled “Effect of obesity and exercise on the expression of the novel myokines, Myonectin and Irisin” submitted by Peterson et al. for publication in PeerJ is a unique work addressing the response of expressional levels of myonectin and FNDC5 in rat diaphragm muscle following chronic exercise training in obese and non-obese rats. This work provides some interesting data, and adds to the current complexity that comprises current myokine research. Although interesting and experimentally sound, this work requires some modification prior to publication.
Title-
The manuscript title suggests that “irisin” expression was measured, but with all of the current controversy surrounding the functional irisin protein, appropriate length, post-translational modification, etc. the word “irisin” should be replaced with “FNDC5”. That being said, if the authors can provide irisin protein levels (as described in the final comment of this review), the title can remain unaltered.

Abstract-
The manuscript addresses change in expressional levels of myonectin and FNDC5 in diaphragm muscle following exercise training in obese and non-obese rats, yet the results portion of the abstract emphasizes too much detail about reference genes in qRT-PCR experiments. The major finding of this work is that myokine expression is elevated in obese versus non-obese rats and that exercise does not seem to stimulate myokine expressional levels (although exercise does increase myonectin protein content). The importance of appropriate reference genes is analytically very important, but the results should be re-written to emphasize the myokine findings rather than the variation of housekeeping genes.

Background-
Page 2 line 73-74- The suggestion that irisin levels may be lower in obese has previously been made, however there is also evidence that suggests circulating irisin is higher in obese subjects (or those with greater BMI) and may correlate most closely with lean body mass. These points should be mentioned in text. Please consider the following papers:
• Polyzos S, Kountouras J, Shields K, Mantzoros C. Irisin: A renaissance in metabolism? Metabolism. 2013; 62: 1037-1044
• Kurdiova T, Balaz M, Vician M, et al. Effects of obesity, diabetes and exercise on Fndc5 gene expression and irisin release in human skeletal muscle and adipose tissue: in vivo and in vitro studies. The Journal of physiology. 2014; 592: 1091-1107
• Huh JY, Panagiotou G, Mougios V, et al. FNDC5 and irisin in humans: I. Predictors of circulating concentrations in serum and plasma and II. mRNA expression and circulating concentrations in response to weight loss and exercise. Metabolism-Clinical and Experimental. 2012; 61: 1725-1738
• Timmons JA, Baar K, Davidsen PK, Atherton PJ. Is irisin a human exercise gene? Nature. 2012; 488: E9-E10

Statistical Page 5, line 181-188- The statistical methods read “A one‐way ANOVA was performed when comparisons were made among all groups (figure 2)” but there is no mention of post-hoc test to control for pair-wise error. The authors should mention which test was performed to control for this (Bonferroni’s, Tukey’s, etc). If not previously performed, one of these techniques should be used prior to publication.

Page 6 line 232- Again, the discussion of circulating irisin in subjects with metabolic disease should be re-written to reflect current data. Also, comparing circulating protein levels in humans and RNA expression of diaphragm muscle of rats is a bit of a translational stretch. Because serum irisin levels were not measured, such comparisons should be made with caution.

Page 6, lines 230-258- The authors mention previous similar rodent training studies and suggest that some of the novelty of the submitted manuscript is the duration of training, specifically that chronic training with consideration for obesity has not been previously performed). The authors should mention major differences between their protocols and those that have been performed by others in references (11, 15) for FNDC5 and reference (12) for myonectin.

Page 6, lines 256-258- The authors speculate that myonectin might possess an autocrine function to control its own expression. It has previously been suggested that myokine expression and secretion may be at regulated in part by NFκB activation and expression, allowing NFκB to bind the promoter regions of common pro-inflammatory myokines which are induced following intense exercise. Is there evidence that myonectin alters NFκB activity? Please consider the following paper:

• Scheler M, Irmler M, Lehr S, et al. Cytokine response of primary human myotubes in an in vitro exercise model. American Journal of Physiology-Cell Physiology. 2013; 305: C877-C886

Results Section- Is there a reason western blots were not performed for diaphragm muscle irisin content but were performed for myonectin? The authors report myonectin gene expression as being reduced however found that protein levels were elevated following training. These experiments should be repeated for FNDC5/irisin because protein (not RNA) is the functional component. These experiments could further add to the novelty of this work and broaden the discussion points. Although it would be interesting to see if serum irisin protein is higher in obese versus non-obese, and if irisin protein is actually unaltered following training, protein levels in diaphragm muscle will still provide some valuable information (if lysates are available).

Reviewer 2 ·

Basic reporting

No Comments

Experimental design

No Comments

Validity of the findings

No Comments

Additional comments

This manuscript investigated the detailed molecular mechanism underlying how exercise stimulates myonectin and irisin/FNDC 5 in LZR and OZR. The results provide compelling evidence that leptin signaling and high fat diet as a potential novel mechanism in the regulation of these proteins (myonectin, FNDC 5). However, there are several issues in the experimental design and the premature discussions in this manuscript.
The authors need to be additional experiments to improve these points and present their data more clearly, which will gain the overall interests from the readers.


Major points:
1. If the authors provide data showing the metabolic status is impaired in an obesity model that induced body weight gain, and insulin resistance, the characteristics of this model should essential include in the result section.
2. The authors cited the ref 16 and17 to explain the using the diaphragm muscle. These studies reported the skeletal function in soleus, gastrocnemius. Please show the physiological relevance of using diaphragm muscle in method and discussion section.
3. The reviewer did not clearly indicate exercise method in cited two papers. They did not match it.
4. Please add data regarding body weight, insulin level and leptin.
5. Regarding the myonectin signaling pathway via irisin in previous studies (ref 23), please explain the underlying the critical results that include the mechanism relation of myonecitin, leptin, and irisin from previous studies.
6. Recently, many studies are arguing the beneficial effect of irisin levels in obese and type 2 disease models during exercise, so the authors may enough to explain the negative regulated by exercise to support these results.
Minor points:
1. Figure 2, and 3: These figures could not distinguish A, B and C.

---

## Round 0.2 · accepted · Accept

In the present study, author reported that long-term exercise training decreases myonectin levels, which is opposite to short-term exercise. These interesting findings confirm earlier work showing that Fndc5 gene expression is not altered by chronic exercise. We are pleased to inform you that your paper has been accepted.